# Cross-sectional study to assess the impact of the COVID-19 pandemic on healthcare services and clinical admissions using statistical analysis and discovering hotspots in three regions of the Greater Toronto Area

Zahra Movahedi Nia [iD] ,[1,2] Cheryl Prescod,[3] Michelle Westin,[3] Patricia Perkins,[2,4] Mary Goitom,[2,5] Kesha Fevrier,[2,6] Sylvia Bawa,[2,7] Jude Kong[1,2]

For numbered affiliations see end of article.

**Correspondence to**
Dr Jude Kong; jdkong@yorku.ca

## ABSTRACT

**Objectives** The COVID-19 pandemic disrupted healthcare services, leading to the cancellation of non-urgent tests, screenings and procedures, a shift towards remote consultations, stalled childhood immunisations and clinic closures which had detrimental effects across the healthcare system. This study investigates the impact of the COVID-19 pandemic on clinical admissions and healthcare quality in the Peel, York and Toronto regions within the Greater Toronto Area (GTA).

**Design** In a cross-sectional study, the negative impact of the pandemic on various healthcare sectors, including preventive and primary care (PPC), the emergency department (ED), alternative level of care (ALC) and imaging, procedures and surgeries is investigated. Study questions include assessing impairments caused by the COVID-19 pandemic and discovering hotspots and critical subregions that require special attention to recover. The measuring technique involves comparing the number of cases during the COVID-19 pandemic with before that, and determining the difference in percentage. Statistical analyses (Mann-Whitney U test, analysis of variance, Dunn's test) is used to evaluate sector-specific changes and inter-relationships.

**Setting** This work uses primary data which were collected by the Black Creek Community Health Centre. The study population was from three regions of GTA, namely, the city of Toronto, York and Peel. For all health sectors, the sample size was large enough to have a statistical power of 0.95 to capture 1% variation in the number of cases during the COVID-19 pandemic compared with before that.

**Results** All sectors experienced a significant decline in patient volume during the pandemic. ALC admissions surged in some areas, while IPS patients faced delays. Surgery waitlists increased by an average of 9.75%, and completed IPS procedures decreased in several subregions.

**Conclusions** The COVID-19 pandemic had a universally negative impact on healthcare sectors across various subregions. Identification of the hardest-hit subregions in

## STRENGTHS AND LIMITATIONS OF THIS STUDY

⇒ Statistical analysis is applied to accurately compare the number of visits, screenings, alternative level of care, waitlists and completed procedures, during the COVID-19 pandemic with before that, in the Greater Toronto Area.

⇒ Critical subregions that require especial attention to recover from the damages caused by the COVID-19 pandemic are extracted and highlighted.

⇒ The study focuses only on the city of Toronto, Peel and York regions of the Greater Toronto Area.

⇒ The dataset is gathered from different sources of only three health clinics, namely, Toronto Public Health Unit (Master No. 3895), York Region Public Health Unit (Master No. 2270) and Peel Public Health Unit (Master No. 2253).

⇒ Since the study is a before-after design, casual relationships are not captured or investigated.

each sector can assist health officials in crafting recovery policies.

## INTRODUCTION

COVID-19 originated in Wuhan, China in December 2019, and quickly spread worldwide through international and intrastate/interstate travel. Shortly thereafter, hospitals and intensive care units (ICUs) were inundated with COVID-19 patients. Many countries implemented lockdown measures to curb the rapid spread of the virus and protect lives. Although non-pharmaceutical interventions (NPI) successfully reduced the number of COVID-19 infections, they had a destructive outcome for the healthcare system.[1] [2] Numerous surgeries, laboratory tests and non-emergency healthcare services

were cancelled or postponed. Additionally, people delayed visiting healthcare centres, fearing COVID-19 infection, and the associated stigma of testing positive for the virus.[3] Consequently, many patients awaiting for healthcare services have fallen out of their timeline, and need to be sorted.

The COVID-19 pandemic had a devastating impact on patients with chronic conditions and non-communicable diseases (NCD), since they were barely able to meet the medical care that they required.[4 5] Equally, mental health disorder increased in adults and children, and adolescents, who due to the situation did not get medical treatment, and therefore, need long-term attention to recover.[6 7] Therefore, it is paramount to identify locations where primary healthcare and emergency department (ED) visits have significantly decreased during the COVID-19 pandemic to provide appropriate medication to individuals that require persistent care and attention.

Another concerning issue that occurred during the COVID-19 pandemic is decline in childhood immunisation. Disruption in children's vaccination, even for a short period of time, could rise the susceptibility to vaccine preventable diseases (VPD) such as measles, polio and pertussis, and increase the risk of VPD outbreaks.[8 9] It is necessary to study this critical matter and provide families with sustained catch-up programmes to rectify missing doses.

Despite urgent need to hospital beds, alternative level of care (ALC) dramatically increased in some places, during the COVID-19 pandemic.[10 11] ALC is the term used to address patients that occupy a hospital bed, but no longer need the intensity of care provided. The most common reasons for the discharge delay in Canada include palliative needs, physical therapy and requiring transition to another adequate facility.[12] ALC is a costly issue which is considered as a bottleneck to hospital facilities, especially during epidemics. It is essential to manage and control ALC at its hotspots for higher efficiency in delivering healthcare services, in future outbreaks.

Studying the impact of the COVID-19 pandemic, and its severity, on the healthcare system and the well-being of patients has become a critical concern to health authorities. The present work aims to assess the effect of COVID-19 pandemic on different health sectors across multiple subregions of the Greater Toronto Area (GTA). Previous studies have examined the adverse effects of the COVID-19 pandemic on various healthcare sectors in other countries/regions. Quaquarini et al[13] conducted a study on the condition of patients with cancer in Lombardy, Italy, compared with previous years. They found that the number of patients with cancer receiving anticancer drug infusions decreased in 2020 compared with 2019 but not compared with previous years. Many patients with cancer experienced treatment delays due to the pandemic, and the number of radiological exams also significantly decreased.

Sutherland et al[14] investigated the impact of the COVID-19 pandemic on a wide range of healthcare procedures. Their results indicated a significant decrease in all activities, including in-person consultations, breast screening, ambulance incidents, ED visits, public hospital inpatients and planned surgeries. Although some recovery was observed by September 2020, a return to normal was still a distant goal. Dorward et al[15] studied the impact of COVID-19 lockdowns on HIV testing and treatment in 65 primary care clinics in KwaZulu-Natal, South Africa. The results revealed that antiretroviral therapy (ART) provisioning remained stable during the lockdowns. However, HIV testing and ART initiations experienced significant declines.

These studies demonstrate that access to medical services across various healthcare sectors decreased in many countries during the pandemic. In the existing scholarly literature, there have been a few attempts to determine the impact of COVID-19 on clinical visits in Ontario. For instance, Ray et al[16] found that the number of hospital and ED care cases related to self-harm, substance use and overdose decreased during the COVID-19 pandemic in Ontario. Conversely, Bouck et al[17] reported an increase in take-home dose coverage for opioid agonist treatment during the pandemic in Ontario.

Data extracted from Public Health Ontario in the work conducted by Mandel et al[18] were used to study the effect of COVID-19 waves on hepatitis B virus and hepatitis C virus testing. The results showed that the number of tests significantly decreased with each wave, although recovering to some extent afterward. Ji et al[19] studied the impact of the COVID-19 pandemic on child immunisation in Ontario. The results indicated a decrease in child immunisation, particularly at 15 and 18 months of age, during the early months of the pandemic, with some recovery later, although not returning to prepandemic levels.

Meggetto et al[20] discovered that cervical screening, colposcopy and treatment volumes significantly decreased in the first 6 months of COVID-19 compared with the pre-COVID-19 pandemic. Vilches et al[21] modelled long-term care facilities in Ontario using an agent-based method. The results suggested that vaccination and regular testing of staff and patients could substantially reduce COVID-19 infections, hospitalisations and deaths. Meaney et al[22] collected a dataset of primary care progress notes in Toronto and applied topic modelling with non-negative matrix factorisation. They used an additive linear model to predict the prevalence of each topic.

Finally, Ferron et al[23] reported a decrease in emergency calls in Niagara, Ontario, during the COVID-19 pandemic. Barrett et al[24] studied the depletion of hospital resources during the COVID-19 pandemic in Ontario, with simulation results indicating that early public health measures could mitigate or delay resource depletion. Additionally, Shoukat et al[25] concluded that self-isolation could significantly reduce the demand for hospital and ICU beds during the COVID-19 pandemic in Canada,

while Saunders *et al*[26] found that mental health services increased among children and adolescents by 10%–15% above expected levels in Ontario during the pandemic.

To further our understanding of the impact of the COVID-19 pandemic on healthcare services in Ontario, we examined various sectors and service departments of healthcare clinics in the city of Toronto, Peel and York regions of the GTA, comparing data from before the pandemic. Our primary focus is on comparing different subregions within this area, a perspective that has not been explored previously. This research is crucial for aiding public health officials and healthcare providers in understanding the impact of the COVID-19 pandemic on different healthcare sectors and for highlighting specific setback hotspots. The findings will be invaluable for informing policies and strategies, guiding future efforts to improve healthcare delivery, mitigating the effects of future pandemics and facilitating a faster recovery and eventual return to the pre-COVID-19 healthcare landscape.

Ontario, with the GTA being a significant contributor, reported the highest number of confirmed COVID-19 cases among all provinces and territories in Canada. However, when adjusted for population size, it ranks sixth due to its large population.[27] Throughout the COVID-19 pandemic (spanning from 2020 to mid-2022), Ontario declared three states of emergency, implementing various levels of restrictions and NPIs, including complete provincial lockdowns.[28] At certain points during the COVID-19 pandemic, many healthcare procedures were significantly impacted, leading to reduced availability of certain services. Non-urgent tests, screenings and procedures were postponed or cancelled to prioritise emergency care and healthcare resources for COVID-19 patients. Many employees and healthcare workers were compelled to work remotely.[29 30] This prolonged disruption significantly strained the healthcare system.

The GTA comprises five distinct regions: Durham, Halton, Peel, York and city of Toronto, and is home to >6 million residents. Approximately 14%, 68% and 18% fall within the age groups below 14, 14–64 and >65 years, respectively. The population is nearly evenly split between males (47%) and females (53%).[31] Most residents seek medical care at local clinics and EDs, rather than hospitals.[32 33] Community health centres play a crucial role in reducing waiting times,[34] addressing geographical healthcare inequalities,[35] serving vulnerable populations[36] and improving the quality of care in local communities.[37 38] The data for this study were collected from three different community clinics.

This paper investigates the impact of the COVID-19 pandemic on the public health system of Toronto and identifies the subregions that have been most affected. Various health sectors, including preventive and primary care (PPC), ED, ALC and imaging, procedure and surgeries (IPS), are compared with data from the pre-COVID-19 period.

## METHODS

This study used data collected by the Black Creek Community Health Centre (BCCHC).[39]

The records used pertain to three distinct health units: Toronto Public Health Unit (Master No. 3895), York Region Public Health Unit (Master No. 2270) and Peel Public Health Unit (Master No. 2253).

The health sectors studied in this work are as follows:

### A. PPC:

► Primary healthcare visits in December 2021 compared with December 2019 (calculated using Equation 1).
► Childhood immunisation in December 2021 compared with December 2019 (calculated using Equation 1).
► Number of faecal, mammogram and pap smear cancer screenings reported during the COVID-19 pandemic compared with January–March 2020 (calculated using Equation 1).

### B. ED:

► Divided into two parts: substance use and mental health. Each part includes the number of visits in December 2021 compared with December 2019.

### C. ALC:

► Number of people in ALC as of 27 March 2022 compared with 31 March 2019 (calculated using Equation 1).

### D. IPS:

► Comprises seven different parts: surgeries, cancer procedures, ophthalmic procedures, orthopaedic procedures, paediatric procedures, MRI imaging and CT scan.
► Three parameters are calculated for each part:
Percentage of patients who have exceeded their time targets.
Completed cases from 28 February to 27 March 2022 compared with the prepandemic period (calculated using Equation 1).
Number of people waiting for surgery as of 21 March 2022 compared with 25 March 2019 (calculated using Equation 1).

In the following sections, all aspects of the study design are thoroughly described.[40]

### Questions to be answered

This work is fundamentally beneficial to the policy makers and health officials working at the GTA to understand the damages caused by the COVID-19 pandemic to the health system, plan for recovery and prepare for future pandemics. Five questions are answered in this work:
1. At the sectors listed below, by what percentage the number of admissions *decreased*, on average, during the

COVID-19 pandemic compared with before that, and where are the hotspots in terms of subregions?

► Primary healthcare
► Childhood immunisation
► Faecal screenings
► Mammogram screenings
► Pap smear screenings
► Substance use
► Mental health

2. By what percentage has ALC *increased*, on average during the COVID-19 pandemic compared with before that, and where are the hotspots in terms of subregions?

3. For the IPS listed below, by what percentage the number of patients who exceeded their time target increased on average, during the COVID-19 pandemic compared with before that, and where are the hotspots of this incident in terms of subregions?

4. For the IPS listed below, by what percentage the completed cases decreased on average, during the COVID-19 pandemic compared with before that, and where are the hotspots in terms of subregions?

► Surgeries
► Cancer procedures
► Ophthalmic procedures
► Orthopaedic procedures
► Paediatric procedures
► MRI imaging
► CT scan

5. By what percentage the number of patients waiting for surgery has *increased* on average, during the COVID-19 pandemic compared with before that, and where are the hotspots in terms of subregions?

## Study population

The data are concentrated on the York and Peel regions, as well as the city of Toronto, encompassing various subregions, including Bramalea, Brampton, Dufferin, East Mississauga, East Toronto, Eastern York Region, Mid-East Toronto, Mid-West Toronto, North Etobicoke Malton West Woodbridge, North Toronto, North York Central, North York West, North West Mississauga, Northern York Region, Scarborough North, Scarborough South, South Etobicoke, South West Mississauga, West Toronto and Western York Region. The data were collected for 151 different forward sortation areas (FSAs). Figure 1 illustrates the study area, with the pink areas representing the FSAs under investigation. The dashed lines, bold black lines and bold red lines demarcate the boundaries of

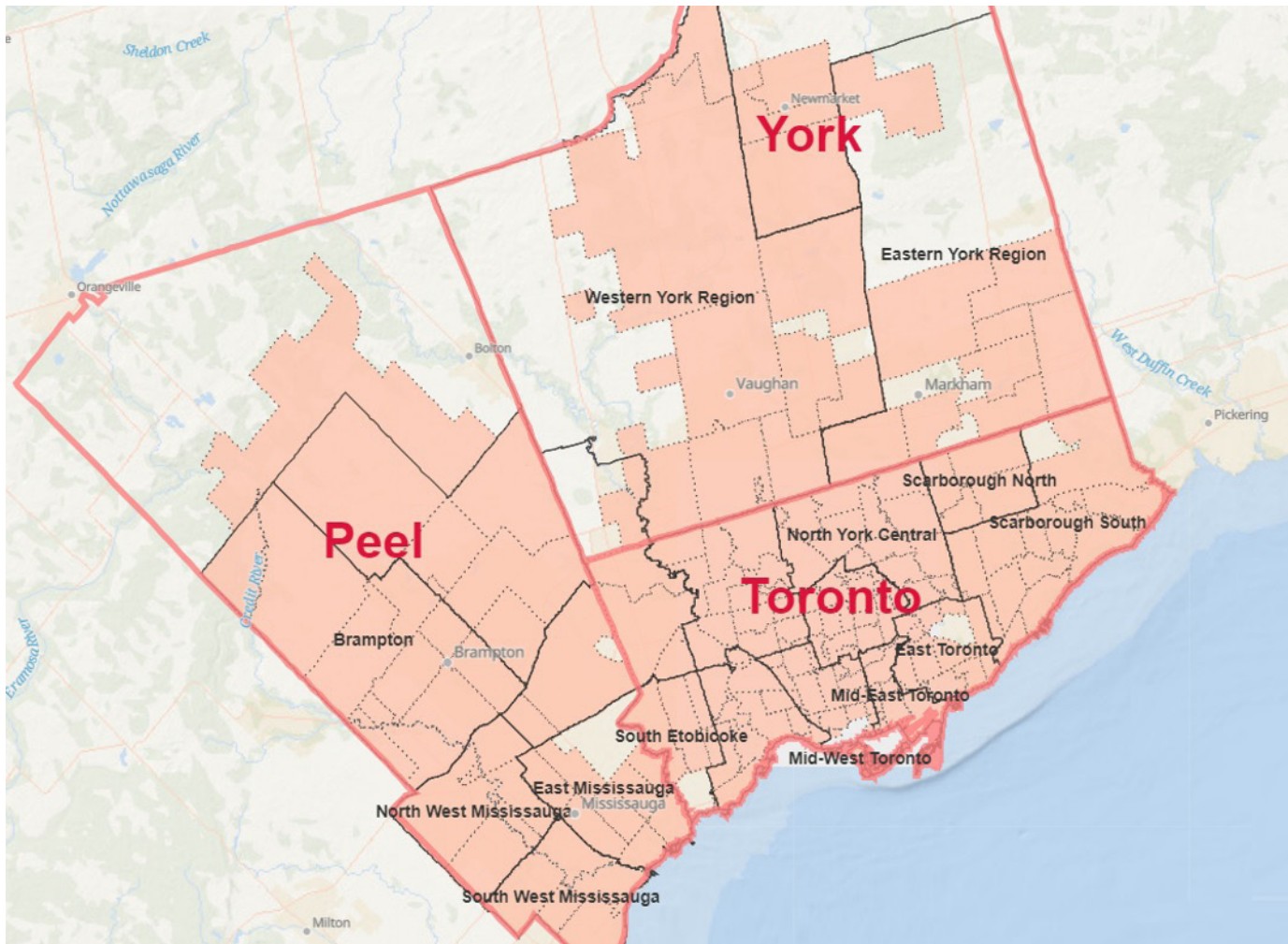

**Figure 1** The area under study.

**Table 1** The population size and percentage of patients with 65 years of age or over

| Subregion | Population | % with 65 years of age or over |
|---|---|---|
| North York West | 806 894 | 14.3768 |
| Eastern York Region | 563 088 | 14.2422 |
| South West Mississauga | 459 340 | 15.8967 |
| East Mississauga | 1 102 439 | 14.8528 |
| North West Mississauga | 1 176 836 | 9.7198 |
| Brampton | 1 497 688 | 9.5578 |
| Bramalea | 1 251 304 | 10.4156 |
| North Etobicoke Malton West Woodbridge | 547 182 | 13.7505 |
| Western York Region | 896 419 | 13.2478 |
| Northern York Region | 206 663 | 12.0321 |
| Dufferin | 8718 | 15.7822 |
| Scarborough South | 448 819 | 14.9264 |
| Scarborough North | 181 127 | 19.9786 |
| East Toronto | 322 882 | 13.3525 |
| North York Central | 372 532 | 16.6389 |
| North Toronto | 245 466 | 15.6355 |
| Mid-East Toronto | 178 652 | 11.0301 |
| Mid-West Toronto | 291 118 | 13.028 |
| West Toronto | 277 071 | 13.434 |
| South Etobicoke | 98 077 | 17.415 |

Population: actual number of patients who have a profile in the health clinics under study, that is, Toronto Public Health Unit (Master No. 3895), York Region Public Health Unit (Master No. 2270) and Peel Public Health Unit (Master No. 2253).
% with 65 years of age or over: percentage of the population that are 65 years old or above.

the FSAs, subregions and regions, respectively. We have created this map using ArcGis Online.[41]

Roughly, half of the population are female (53%). Patients are from all age groups. Table 1 shows the total population in each subregion, and the percentage of people that are 65 years old or above.

### Type of study and unit of observation

This study consumes primary data and is considered as an observational epidemiological study, and a cross-sectional study. The unit of analysis for primary healthcare, cancer screenings (faecal, mammogram and pap smear) and ED (substance use and mental health) is the number of visits, and for childhood immunisation, ALC and IPS sector (patients who have exceeded their time target, patients who have completed their IPS and patients waiting for surgery) is the number of patients.

### Measuring technique

The data were collected from various sources, including the Ontario Health Insurance Plan (OHIP), Client and Health Related Information System, Transfer Payment Ontario, Ontario Drug Benefit Claims, Drug and Alcohol Treatment Information System, Narcotics Monitoring System, Bed Census Summary, Ontario Healthcare Financial and Statistics, Wait Time Information System, Resource Matching and Referral, Home and Community Care Support Services, among others.

For certain parameters within the dataset, we compared the number of patients during the COVID-19 pandemic to the period before that for each subregion, using Equation 1.

$$100 \times \left( \frac{patients-during-COVID-19}{patients-before-COVID-19} - 1 \right) \quad (1)$$

Where, $patients-during-COVID-19$ and $patients-before-COVID-19$ are the number of patients during and before COVID-19 periods, respectively. So, if the number of patients has increased during COVID-19 compared with before that, Equation 1 will return a positive value and if it has decreased, it will return a negative value.

Missing data which happened only in the PPC sector due to unavailability of the location variable in the OHIP data source, accounted for approximately 12% of the PPC data, and was excluded from the dataset.

The health parameters of each health sector were compared across different subregions using histograms, box and violin plots and through the Mann-Whitney U test. To identify which subregion is significantly different from the rest, first, analysis of variance (ANOVA) was used to determine the significance, and then Dunn's test with Bonferroni correction was applied to identify the exact subregion(s). P values <0.05 are considered significant.[42] This study uses the Strengthening the Reporting of Observational Studies in Epidemiology cross-sectional reporting guidelines.[43]

### Calculation of sample size

The sample size of the study was set large enough to have a statistical power of 0.95 for capturing 1% variation in the number of cases, during COVID-19 pandemic compared with before that, for all health parameters under study.[44] Table 1 illustrates the total population size for different subregions.

### Patient and public involvement

Patients or the public were not involved in the design, or reporting, or dissemination plans of this research.

### RESULTS
### Preventive and primary care

PPC includes health parameters such as primary health-care visits, childhood immunisation compared with December 2019, and cancer screening compared with

**Table 2** Reduction of different PPC sectors during versus before the COVID-19 pandemic

| PPC | Average change | 95% CI | ANOVA p value | Subregion with fewer patients |
|---|---|---|---|---|
| Primary healthcare visits | −3.39% | Min. −4.14% Max. −2.64% | $1.39 \times 10^{-7}$ | Scarborough South, East Toronto and Scarborough North |
| Childhood immunisation | −8.56% | Min. −11.1% Max. −6.01% | 0.017 | Scarborough South |
| Faecal cancer screening | 28.77% | Min. 26.68% Max. 30.86% | 0.026 | Mid-East Toronto and North York West |
| Mammogram screening | −1.33% | Min. −3.82% Max. −1.16% | $2.41 \times 10^{-7}$ | East Mississauga, Scarborough South and South Etobicoke |
| Pap smear screening | 29.09% | Min. 26.56% Max. 31.62% | 0.074 | Mid-East Toronto and North York West |

Average change: negative/positive values indicate that the number of patients has decreased/increased on average.
95% CI: negative/positive values indicate that the lower (min.) or upper (max.) bounds for the 95% CI is a decrease/increase compared with before the COVID-19 pandemic.
ANOVA p value: any value <0.05 indicates that at least one location is significantly different form the rest. Subregion with fewer patients: extracts the critical subregions that need special attention to recover.
ANOVA, analysis of variance; max., maximum; min., minimum; PPC, preventive and primary care.

January–March 2020, which are presented below. The results are summarised in table 2.

### Primary healthcare visits

We compared the number of primary care visits in December 2021 with those in 2019. During the pandemic, the number of visits experienced an average decrease of 3.39% (with a 95% CI ranging from a minimum of 2.64% to a maximum of 4.14%). The histogram (online supplemental figure 1(a)) and distribution (online supplemental figure 1(b)) of primary healthcare visits during versus before COVID-19 reveals that only the Northern York Region did not experience a decrease in the number of visits during the pandemic. Moreover, the North York Central, North Toronto and Northeast Toronto subregions suffered less compared with other areas. Furthermore, approximately half of the visits were not conducted in person, as shown in online supplemental figure 2.

The ANOVA p value for primary healthcare visits in December 2021 compared with December 2019, and the percentage of in-person visits, was $1.39 \times 10^{-7}$ and $3.03 \times 10^{-9}$, respectively, indicating that in both parameters, a subregion is significantly different from the rest. The p values matrix of Dunn's test (online supplemental figure 3(a and b)) shows that the number of primary healthcare visits in December 2021 has significantly decreased in Scarborough South, East Toronto and Scarborough North, compared with December 2019 (table 2), and the number of in-person visits was higher in North York West, Northern York Region and Scarborough South.

### Childhood immunisation compared with December 2019

The average childhood immunisation has decreased >50% in some subregions, in December 2021 compared with December 2019 (online supplemental figure 4(a)).

However, due to a substantial increase (up to >150%) in some areas of North, Mid-East and Mid-West Toronto, childhood immunisation has decreased by only about 8.56% on average (with a 95% CI ranging from a minimum of 6.01% to a maximum of 11.1%) compared with the pre-COVID-19 period.

The ANOVA p value is 0.017, indicating that the change in childhood immunisation during COVID-19, compared with the period before, was significantly different for different subregions. Using Dunn's test (online supplemental figure 4(b)), we observed that childhood immunisation has reduced more in Scarborough South compared with before the COVID-19 pandemic. Table 2 summarises the results.

### Cancer screening compared with January–March 2020

The number of three different types of cancer screenings, namely, faecal, mammogram and pap smear, during the COVID-19 pandemic has been compared with the prepandemic period. Online supplemental figure 5(a and b) presents a comparison of their histograms, and distribution across different subregions, respectively. It is evident from online supplemental figure 5 that among the three types, mammogram screenings experienced the most significant decrease. The results of the Mann-Whitney U test (online supplemental figure 5(b)) also indicate that the distribution of faecal and pap smear cancer screenings is closer to each other than to mammogram screenings. Specifically, the number of faecal and pap smear screenings increased by about 28.77% (with a 95% CI ranging from a minimum of 26.68% to a maximum of 30.86%) and 29.09% (with a 95% CI ranging from a minimum of 26.56% to a maximum of 31.62%) on average, respectively, during the COVID-19 pandemic compared with

the pre-COVID-19 period. In contrast, the number of mammogram screenings decreased by 1.33% (with a 95% CI ranging from a minimum of 1.16% to a maximum of 3.82%) on average during the same period. Among all the different subregions, East Mississauga, Scarborough South and South Etobicoke have shown a decrease in the number of mammogram tests compared with the period before COVID-19.

The ANOVA test for the reduction in faecal, mammogram and pap smear screening during COVID-19, compared with prepandemic levels yielded p values of 0.026, $2.41 \times 10^{-7}$ and 0.074, respectively. This suggests that the reduction in the number of cancer screenings, when compared with the period before the COVID-19 pandemic is significantly different in different subregions for mammogram, and then for faecal tests. However, the reduction in pap smear tests is approximately the same for the different subregions in the case of pap smear screenings.

According to Dunn's test (online supplemental figure 6(a, b and c)), Mid-East Toronto and North York West have experienced the most significant reduction in faecal and pap smear screenings compared with the period before the COVID-19 pandemic, and East Mississauga, Scarborough South and South Etobicoke for mammogram tests. This has also been summarised in table 2.

### Emergency department

Substance use and mental health were studied under the category of ED. The average rates of substance use and mental health visits in December 2021 are 419.17 and 877.24 per 100 000 people, respectively. Both substance use and mental health visits had significantly decreased in December 2021 compared with December 2019 (online supplemental figure 7(a and b)).

On average, the number of substance use visits and mental health visits has decreased by 2.25% (with a 95% CI ranging from a minimum of −7.22% to a maximum of 11.73%) and 2.25% (with a 95% CI ranging from a minimum of −4.8% to a maximum of 9.29%) in

December 2021 compared with December 2019, respectively. The ANOVA test for substance use and mental health across different subregions yielded p values of 0.52 and 0.014, respectively. This suggests that the reduction in the number of visits, compared with before COVID-19, is approximately the same for different subregions for substance use, and significantly different for mental health. Moreover, Dunn's test shows that the reduction in substance use visits is more significant in Brampton, and the reduction in mental health visits is more significant in South West Mississauga and West Toronto (refer to online supplemental figure 8(a and b)). Table 3 summarises the results of ED and ALC.

### Alternative level of care

ALC as of 27 March 2022 decreased by 33.66 (minimum and maximum of −15.71 and 15.04 with 95% CI) percentage compared with 31 March 2019, on average. However, the histogram (online supplemental figure 9(a)) shows that ALC has increased by >200% in some FSAs compared with pre-COVID-19 period. The violin plot (online supplemental figure 9(b)) shows that ALC dramatically increased by >200% in Western York Region, Scarborough South and Mid-East Toronto. The ANOVA p value for ALC is 0.76 and indicates that none of the subregions are significantly different from the other. The Dunn's test (online supplemental figure 9(c)) indicates that there is no hotspot as well. A summary of the results is provided in table 3.

### Imaging, procedures and surgeries

Approximately half of the patients waiting for various procedures exceeded their timeline targets (as shown in online supplemental figure 10(a)). On average, the number of patients that have fallen out of their time targets for surgeries equals to 53.84% (with a 95% CI ranging from a minimum of 53.11% to a maximum of 54.56%), for cancer tests comes to 48.52% (with a 95% CI ranging from a minimum of 46.53% to a maximum of 48.29%), for ophthalmology appointments corresponds

**Table 3** Change in number of patients of ED and ALC during the COVID-19 pandemic compared with before that

| Health sector | Average change | 95% CI | ANOVA p value | Subregion with fewer patients |
|---|---|---|---|---|
| Substance use | −2.25% | Min. −7.22% Max. 11.73% | 0.52 | Brampton |
| Mental health | −2.2.5% | Min. −4.8% Max. 9.29% | 0.014 | South West Mississauga and West Toronto |
| | | | | Subregion with more patients |
| ALC | −33.66% | Min. −15.71% Max. 15.04% | 0.76 | Western York Region, Scarborough South and Mid-East Toronto |

Average change: negative/positive values indicate that the number of patients has decreased/increased on average.
95% CI: negative/positive values indicate that the lower (min.) or upper (max.) bounds for the 95% CI is a decrease/increase compared with before the COVID-19 pandemic.
ANOVA p value: any value <0.05 indicates that at least one location is significantly different form the rest. Subregion with fewer patients: extracts the critical subregions that need special attention to recover. No values, '−' indicate that no subregion could be extracted, because the suffer is roughly the same for all subregions.
ALC, alternative level of care; ANOVA, analysis of variance; ED, emergency department; max., maximum; min., minimum.

**Table 4** ANOVA and Dunn's test for different IPS parameters

| IPS | Average change | 95% CI | ANOVA p value | Subregion with more patients |
|---|---|---|---|---|
| Surgeries | 53.84% | Min. 53.11% Max. 54.56% | $6.96 \times 10^{-13}$ | Brampton, Bramalea, Mid-West Toronto |
| Cancer procedures | 48.52% | Min. 46.53% Max. 50.51% | 0.091 | Scarborough South |
| Ophthalmic procedures | 47.27% | Min. 46.26% Max. 48.29% | $2.22 \times 10^{-6}$ | Northern York Region |
| Orthopaedic procedures | 54.86% | Min. 53.54% Max. 56.18% | $1.88 \times 10^{-16}$ | Brampton, Bramalea |
| Paediatric procedures | 59.27% | Min. 58.04% Max. 60.51% | 0.014 | North Etobicoke Malton West Woodbridge, East Toronto |
| MRI | 60.8% | Min. 59.7% Max. 61.83% | $6.92 \times 10^{-35}$ | North Toronto, Mid-East Toronto, Mid-West Toronto |
| CT scan | 61.84% | Min. 60.97% Max. 62.71% | $2.58 \times 10^{-22}$ | East Toronto, North Toronto, Mid-East Toronto |

Average change: positive values indicate that how much has the number of patients waiting for a procedure has increased on average.
95% CI: positive values indicate that the lower (min.) or upper (max.) bounds for the 95% CI is an increase compared with before the COVID-19 pandemic.
ANOVA p value: any value <0.05 indicates that at least one location is significantly different form the rest. Subregion with more patients: extracts the critical subregions that need special attention to recover. Because the number of their patients waiting for service has increased more compared with other locations.
ANOVA, analysis of variance; IPS, imaging, procedures and surgeries; max., maximum; min., minimum.

to 47.27% (with a 95% CI ranging from a minimum of 53.54% to a maximum of 56.18%), for orthopaedic procedures sits at 54.86% (with a 95% CI ranging from a minimum of 58.04% to a maximum of 60.51%), for paediatrics is 59.27% (with a 95% CI ranging from a minimum of 58.04% to a maximum of 60.51%), for MRI scans agrees with 60.8% (with a 95% CI ranging from a minimum of 59.7% to a maximum of 61.83%) and for CT scans matches up with 61.84% (with a 95% CI ranging from a minimum 60.97% to a maximum of 62.71%). Please refer to table 4 and online supplemental figure 10(b–h) for the ANOVA and Dunn's test results for IPS waitlists.

Online supplemental figure 11(a and b) display the histogram and distribution of the number of patients waiting for surgery during the COVID-19 pandemic compared with prepandemic levels. On average, the number of patients waiting for surgery increased by 9.75% (with a 95% CI ranging from a minimum of 7.95% to a maximum of 11.55%) during the COVID-19 pandemic compared with the prepandemic period.

Surgery procedures predominantly increased in various subregions during the COVID-19 pandemic compared with before the pandemic (online supplemental figure 11(a and b)). Subregions such as Eastern York, South West Mississauga, North Etobicoke Malton West Woodbridge, Bramalea, Northern York, Scarborough South, Scarborough North, Mid-East Toronto and South Etobicoke have recorded an increase in the number of patients waiting for surgery during the COVID-19 pandemic. The ANOVA p value for changes in the number of people

waiting for surgery compared with before the COVID-19 pandemic is equal to $8.95 \times 10^{-12}$. Dunn's test results (refer to online supplemental figure 11(c)) indicate that the increase in the number of patients on surgical waitlists is more pronounced in Scarborough South, Northern York Region and Bramalea compared with the pre-COVID-19 period.

As shown in online supplemental figure 12(a and b), on average, the number of completed procedures for cancer tests, MRI scans and CT scans increased by 7.15% (with a 95% CI ranging from a minimum of −0.14% to a maximum of 0.14%), 0.58% (with a 95% CI ranging from a minimum of −1.16% to a maximum of 2.78%) and 2.38% (with a 95% CI ranging from a minimum of 0.33% to a maximum of 4.43%), respectively. However, for surgeries, ophthalmology, orthopaedics and paediatrics, the number of completed procedures decreased by 16.07% (with a 95% CI ranging from a minimum of 13.73% to a maximum of 18.41%), 11.12% (with a 95% CI ranging from a minimum of 6.76% to a maximum of 15.48%), 4.44% (with a 95% CI ranging from a minimum of −0.26% to a maximum of 9.15%) and 24.82% (with a 95% CI ranging from a minimum of 18.96% to a maximum of 30.67%) during the COVID-19 pandemic compared with before, respectively. The results are summarised in table 5.

Approximately, in half of the locations, the number of completed procedures for cancer tests, MRI scans and CT scans increased during the COVID-19 pandemic (online supplemental figure 12(a)). In addition, in a few locations, the number of completed cancer tests increased

**Table 5** ANOVA and Dunn's test results for completed IPS during COVID-19 compared with prepandemic levels

| IPS | Average change | 95% CI | ANOVA p value | Subregion with less completed cases |
|---|---|---|---|---|
| Surgeries | −16.07% | Min. −18.41% Max. −13.73% | 0.50 | – |
| Cancer procedures | 7.15% | Min. −0.14% Max. 0.14% | 0.43 | – |
| Ophthalmic procedures | −11.12% | Min. −15.48% Max. −6.76% | 0.20 | East Mississauga |
| Orthopaedic procedures | −4.44% | Min. −9.15% Max. −0.26% | 0.37 | – |
| Paediatric procedures | −24.82% | Min. −30.67% Max. −18.96% | 0.20 | – |
| MRI | 0.58% | Min. −1.16% Max. 2.78% | 0.00311 | North York Central, Mid-East Toronto, West Toronto |
| CT scan | 2.38% | Min. 0.33% Max. 4.43% | 0.00429 | Scarborough South, Scarborough North |

Average change: positive/negative values indicate that less/more procedures have completed compared with before the COVID-19 pandemic.
95% CI: positive/negative values indicate that the lower (min.) or upper (max.) bounds for the 95% CI is an increase/decrease compared with before the COVID-19 pandemic.
ANOVA p value: any value <0.05 indicates that at least one location is significantly different form the rest. Subregion with more patients: extracts the critical subregions that need special attention to recover. Because the completed procedures in them have decreased more compared with before the pandemic.
ANOVA, analysis of variance; IPS, imaging, procedures and surgeries; max., maximum; min., minimum.

by >150%, contributing to the average increase of 7.15% (online supplemental figure 12(b)). Moreover, both figures show that the number of completed surgeries and paediatric procedures significantly decreased during the pandemic compared with before.

Table 5 reveals that for most IPS, the ANOVA p value is high across subregions, and for a few of them, Dunn's test identifies specific subregions with a significant decrease in the number of completed cases during COVID-19 (online supplemental figures 13 and 14).

Interestingly, the results indicate that although the number of completed CT scans has increased by 2.38% on average during the COVID-19 pandemic compared with before (table 5), the number of patients who have surpassed their timeline increased the most for CT scan among all other ISP, according to table 4. The reason lies in the high demand for CT scan during the pandemic to classify lung injury in patients diagnosed with COVID-19. Moreover, angio-CT was frequently used during the pandemic to confirm or inform suspicious pulmonary embolism in severe COVID-19 cases.[45]

## DISCUSSION

This study suggests that in all clinical sectors, including PPC and ED, the number of visits significantly decreased during the COVID-19 pandemic compared with before that. For each health sector, the Dunn's test was used to extract the regions which suffered the most. The number of visits to the PPC sector decreased the most in Scarborough South, East Toronto and Scarborough North. This disparity was also supported by the ANOVA test. Additionally, childhood immunisation was reduced more in Scarborough South. Faecal and pap-smear screenings decreased more in Mid-East Toronto and North York West, and mammogram screenings declined the most in Scarborough South and South Etobicoke which was also supported by the ANOVA test. From the ED, substance use visits decreased in Brampton, and mental health visits were less in South West Mississauga. ALC appeared to be very high in three subregions, Western York Region, Scarborough South and Mid-East Toronto. The waitlist increased the most in Brampton, Bramalea and Mid-West Toronto for surgeries, in Scarborough South for cancer procedures, in Northern York region for ophthalmic procedures, in Brampton and Bramalea for orthopaedic procedures, in North Etobicoke Malton West Woodbridge and East Toronto for paediatric procedures, in North, Mid-East and Mid-West for MRI and in East, North and Mid-East Toronto for CT scan. Moreover, less procedures were completed in East Mississauga for ophthalmic, in North York Central, Mid-East and West Toronto for MRI, and in Scarborough South and Scarborough North for CT scan.

Our findings could be generalised to health services in other locations, since other studies have also found similar pattern of incidences in other countries. Pescariu et al[46] found that intracardiac device implantation procedures decreased by 75% during the COVID-19 pandemic, in Romania. Quaquarini et al[13] reported a significant decline in radiological exams during the COVID-19 pandemic,

in Lombardy, Italy. Sutherland *et al*[14] found that during the COVID-19 pandemic, breast screening decreased by 51.5%, ED visits by 13.9% and hospital planned surgeries by 32.6% compared with before, in Australia.

Moreover, other studies found similar results in Ontario.[16 19 20 23] However, our work is novel since it has studied the impact of COVID-19 on healthcare system in a more granular scale, that is, subregions, to extract critical localities that require special attention for recovery.

The COVID-19 pandemic has had a negative impact on all the health sectors under study. Patients with high acuity conditions such as chronic disease and NCD, mental health disorder and patients that need urgent IPS services, that have postponed or missed their medication need to be ministered immediately, before they are too ill to be treated adequately. For some conditions, such as cancer, long waiting times increase the risk of developing advanced cancer and metastases. Therefore, based on the criticality of the condition, patients must be prioritised and recovery programmes need to be informed.

For children that have missed their VPD vaccine doses, some monitoring and compensation programme is necessary to refrain preventable outbreaks. Increased public awareness regarding outcomes of child immunisation negligence could also help with recovery as it elevates public cooperation. Additionally, it will be profoundly beneficial to identify localities where high volumes of ALC occur, and inform ALC avoidance policies for improving healthcare efficiency. Furthermore, it is worth mentioning that the trauma caused by the COVID-19 pandemic may have long-term consequences for the patients and healthcare system that is not possible to measure or understand yet.

This study has many strengths and some limitations. In this work, statistical analysis has been used to accurately compare the number of visits, waitlists and completed procedures during the COVID-19 pandemic with before that. In addition, subregions that have suffered more in the Peel and York regions, and the city of Toronto have been highlighted for each health sector. This could be highly beneficial as it lightens the performance of different health sectors at subregion level. However, the data were driven only from three health clinics, and from limited number of sources. Moreover, since the study has a before-after design, casual relationships are not captured and cannot be investigated.

### Implications for policy and practice

To mitigate the spread of the virus and protect people's lives, many non-essential care services were cancelled or postponed. Furthermore, the availability of healthcare personnel was reduced, with many healthcare providers working remotely. Additionally, patients were hesitant to visit doctors due to fear of the COVID-19 and the associated stigma of testing positive for the virus. To address these challenges, additional measures need to be implemented to compensate for the reduction in clinical visits. For example, continuous monitoring is necessary

for children who have not been immunised, and proper care must be provided for patients with cancer and ED patients who have not been tested or treated.

While ALC has decreased in most areas, it has increased by up to 200% in some subregions during the COVID-19 pandemic compared with before that. It is essential to expedite the discharge of ALC patients and ensure that services are available to accommodate the necessary care after the pandemic. The decrease in the number of surgeries and diagnostic procedures during the COVID-19 pandemic resulted in many patients exceeding their waiting time limits. Additionally, the number of completed procedures significantly declined in various subregions during the pandemic compared with the period before. Immediate actions are needed to manage surgeries and diagnostic services effectively to avoid negative health outcomes for those with long-term and life-threatening illnesses.

### CONCLUSION

The COVID-19 pandemic has had a devastating impact on various sectors of the healthcare system. This study compares the performance of different healthcare sectors with the pre-COVID-19 era and identifies the subregions that were most affected. Identifying these hotspots of setbacks in each healthcare sector can assist health officials and care providers in focusing their efforts on those areas for a faster recovery. This research specifically focuses on the subregions of Peel, York and the city of Toronto. It is worth noting that other areas within the GTA may be studied in the future.

**Author affiliations**
[1]Africa-Canada Artificial Intelligence and Data Innovation Consortium (ACADIC), Department of Mathematics and Statistics, York University, Toronto, Ontario, Canada
[2]Resilience Research Atlantic Alliance on Sustainability, Supporting Recovery and Renewal (REASURE2) Network, Toronto, Ontario, Canada
[3]Black Creek Community Health Centre, Toronto, Ontario, Canada
[4]Faculty of Environment and Urban Change, York University, Toronto, Ontario, Canada
[5]School of Social Work, York University, Toronto, Ontario, Canada
[6]Department of Geography and Planning, Queen's University, Kingston, New York, Canada
[7]Department of Sociology, York University, Toronto, Ontario, Canada

**Contributors** ZMN: conceptualisation, investigation, methodology, software, validation, visualisation, writing—original draft, writing—review and editing. CP: conceptualisation, data curation, resources, investigation, formal analysis, writing—review and editing. MW: conceptualisation, data curation, resources, investigation, formal analysis, writing—review and editing. PP, MG: conceptualisation, funding acquisition, formal analysis, validation, writing—review and editing. KF, SB: conceptualisation, funding acquisition, formal analysis, validation, writing—review and editing. JK: conceptualisation, data curation, formal analysis, funding acquisition, investigation, methodology, project administration, software, supervision, validation, writing—original draft, writing—review and editing, and the guarantor of the manuscript. All authors approve the final version of the manuscript. All authors agree to be accountable for all aspects of the work in ensuring that questions related to the accuracy or integrity of any part of the work are appropriately investigated and resolved.

**Funding** This research is funded by the Social Sciences and Humanities Research Council (SSHRC)-New Frontier in Research Fund-Exploratory (grant no. NFRFE-2021-00879). JK acknowledges support from Canada's International Development

Research Centre (IDRC) (grant no. 109981), support from NSERC Discovery Grant (grant no. RGPIN-2022-04559) and NSERC Discovery Launch Supplement (grant no. DGECR-2022-00454).

**Map disclaimer** The inclusion of any map (including the depiction of any boundaries therein), or of any geographic or locational reference, does not imply the expression of any opinion whatsoever on the part of BMJ concerning the legal status of any country, territory, jurisdiction or area or of its authorities. Any such expression remains solely that of the relevant source and is not endorsed by BMJ. Maps are provided without any warranty of any kind, either express or implied.

**Competing interests** None declared.

**Patient and public involvement** Patients and/or the public were not involved in the design, or conduct, or reporting, or dissemination plans of this research.

**Patient consent for publication** Not applicable.

**Ethics approval** Not applicable.

**Provenance and peer review** Not commissioned; externally peer reviewed.

**Data availability statement** Data are available on reasonable request. The dataset of this work is available from the corresponding author on a reasonable request with the permission of the Black Creek Community Health Centre (BCCHC).

**ORCID iD**
Zahra Movahedi Nia http://orcid.org/0000-0002-5528-638X

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
