## [Reviewer comments · BMJ Open]

ARTICLE DETAILS

TITLE (PROVISIONAL)	A cross-sectional study to assess the impact of the COVID-19 pandemic on healthcare services and clinical admissions using statistical analysis and discovering hotspots in three regions of the Greater Toronto Area
AUTHORS	Movahedi Nia, Zahra; Prescod, Cheryl; Westin, Michelle; Perkins, Patricia; Goitom, Mary; Fevrier, Kesha; Bawa, Sylvia; Kong, Jude

VERSION 1 – REVIEW

REVIEWER	Swarnakar, Raktim All India Institute of Medical Sciences, physical medicine and rehabilitation
REVIEW RETURNED	04-Jan-2024

GENERAL COMMENTS	Dear authors, 1. Please discuss in terms of Generalisability of the study in the discussion section. 2. What is the implications and new things of this study in the background of all available similar studies. Thank you.
---

REVIEWER	Tudoran, Cristina University of Medicine and Pharmacy Victor Babes Timisoara, Department VII. Internal Medicine II, Cardiology Clinic I
REVIEW RETURNED	04-Jan-2024

GENERAL COMMENTS	This is an interesting and well-structured article. I have some suggestions to the authors. 1. The fact that the authors mention only the number of the reference without the name of the first author, makes the manuscript difficult to read: "In [17], researchers" – line 12., etc. 2. What does "3895 Toronto" at line 31, page 4, mean? Is it the number of patients? 3. The manuscript would benefit if the authors would use a subtitle to highlight the statistical analyse. 4. The authors could write a legend for each table. 5. The sentence " On average, 53.84% (with a 95% CI ranging from a minimum of 53.11% to a maximum of 54.56%), 48.52% (with a 95% CI ranging from a minimum of 46.53% to a aximum of 50.51%), 47.27% (with a 95% CI ranging from a minimum of 46.26% to a maximum of 8.29%), 54.86% (with a 95% CI ranging from a minimum of 53.54% to a maximum of 56.18%), 59.27% (with a 95% CI ranging from a minimum of 58.04% to a maximum of 60.51%), 60.8% (with a 95% CI ranging from a minimum of 59.7% to a maximum of 61.83%), and 61.84% (with a 95% CI ranging from a minimum of 60.97% to a maximum of 62.71%) of
--

	patients waiting for surgery, cancer tests, ophthalmology appointments, orthopedic procedures, pediatrics, MRI scans, and CT scans, respectively, have exceeded their waiting time targets." is difficult to understand and should be rephrased; page7, line 53. 6. " Approximately half of the locations, the number of" should be rephrased, page 9, line 15. 7. In Discussion the authors mention that the number of CT scans decreased. I suggest to debate more on this affirmation, as the number of lung CT scans was influenced by the need to classify the lung injury in patients diagnosed with COVID-19 and angio-CT was frequently used to confirm or inform a suspicion of pulmonary embolism, with a high incidence in hospitalized, sever COVID-19 cases, as stated in Tudoran Cristina, Dana Emilia Velimirovici, Delia Mira Berceanu-Vaduva, Maria Rada, Florica Voiță-Mekeres, and Mariana Tudoran. 2022. "Increased Susceptibility for Thromboembolic Events versus High Bleeding Risk Associated with COVID-19" Microorganisms 10, no. 9: 1738. https://doi.org/10.3390/microorganisms10091738 8. Overall, the Discussion section is too short. The study would benefit if the authors would compare their findings with research from other countries, for example: Pescariu Silviu Alexandru, Cristina Tudoran, Gheorghe Nicusor Pop, Sorin Pescariu, Romulus Zorin Timar, and Mariana Tudoran. 2021. "Impact of COVID-19 Pandemic on the Implantation of Intra-Cardiac Devices in Diabetic and Non-Diabetic Patients in the Western of Romania" Medicina 57, no. 5: 441. https://doi.org/10.3390/medicina57050441 9. Usually Limitations is at the end of Discussions.
--	---

REVIEWER	Kaneko, Makoto Yokohama City University, Primary Care Research Unit, Graduate School of Health Data Science
REVIEW RETURNED	18-Jan-2024

GENERAL COMMENTS	To the authors Thank you for providing the opportunity to review the article. My below comments are based on the STROBE checklist for observational study (http://www.equator-network.org/reporting-guidelines/strobe/) and I hope these comments improve your article. Major concerns Title and abstract #1. Please include the study's design with a commonly used term in the title or the abstract. Introduction #2. The introduction is slightly redundant and the author merely lists up existing research regarding the impact of COVID-19. Thus, please rewrite the section concisely. Methods #3. What does ALC (Alternative Level of Care) mean? #4. Please describe the study design. Results
---

	#5. Please demonstrate the proportion of missing data among all data #6. The study is a before-after design. Thus, the authors can not investigate causal relationships. This needs to be included in the limitation section. Minor concerns #7. In the Methods section, the authors stated the STROB. It may indicate STROBE.
--	---

VERSION 1 – AUTHOR RESPONSE

Reviewer: 1

Dr. Raktim Swarnakar, All India Institute of Medical Sciences

Comments to the Author:

Dear authors,

1. Please discuss in terms of Generalisability of the study in the discussion section.

Response:

Thank you very much for this important comment. We have compared our results with the results from other countries in the discussion to discuss generalisability of our work:

Our findings could be generalized to health services in other locations, since other studies have also found similar pattern of incidences in other countries. Pescariu, et al [46] found that intra-cardiac device implantation procedures decreased by 75% during the COVID-19 pandemic, in Romania. Quaquarini, et al [13] reported a significant decline in radiological exams during the COVID-19 pandemic, in Lombardy, Italy. Sutherland, et al [14] found that during the COVID-19 pandemic, breast screening decreased by 51.5%, emergency department visits by 13.9%, and hospital planned surgeries by 32.6% compared to before, in Australia.

Moreover, other studies found similar results in Ontario [16, 19, 20, 23].

2. What is the implications and new things of this study in the background of all available similar studies.

Response:

Thank you very much for this great comment. We have discussed the novelty and implications of our study in the discussion:

Moreover, other studies found similar results in Ontario [16, 19, 20, 23]. However, our work is novel since it has studied the impact of COVID-19 on healthcare system in a more granular scale, i.e., subregions, to extract critical localities that require especial attention for recovery.

The COVID-19 pandemic has had a negative impact on all the health sectors under study. Patients with high acuity conditions such as chronic and non-communicable disease, mental health disorder, and patients that need urgent IPS services, that have postponed or missed their medication need to be ministered immediately, before they are too ill to be treated adequately. For some conditions, such as cancer, long waiting times increase the risk of developing advanced cancer and metastases. Therefore, based on the criticality of the condition, patients must be prioritized and recovery programs need to be informed.

For children that have missed their VPD vaccine doses, some monitoring and compensation program is necessary to refrain preventable outbreaks. Increased public awareness regarding outcomes of child immunization negligence could also help with recovery as it elevates public cooperation. Additionally, it will be profoundly beneficial to identify localities where high volumes of ALC occur, and inform ALC avoidance policies for improving health care efficiency. Furthermore, it is worth mentioning that the trauma caused by the COVID-19 pandemic may have long-term consequences for the patients and healthcare system that is not possible to measure or understand yet.

Reviewer: 2

Dr. Cristina Tudoran, University of Medicine and Pharmacy Victor Babes Timisoara

Comments to the Author:

This is an interesting and well-structured article.

I have some suggestions to the authors.

1. The fact that the authors mention only the number of the reference without the name of the first author, makes the manuscript difficult to read: "In [17], researchers" – line 12., etc.

Response:

Thank you very much for your helpful comment. We have revised the manuscript and added the first author's name to all of the previous works that are reviewed.

2. What does "3895 Toronto" at line 31, page 4, mean? Is it the number of patients?

Response:

Thank you very much for your attention. 3895 is the master number of Toronto public health Unit. We have clarified this in our manuscript to fix this concern.

- Toronto Public Health Unit (Master No. 3895)
- York Region Public Health Unit (Master No. 2270)
- Peel Public Health Unit (Master No. 2253)

3. The manuscript would benefit if the authors would use a subtitle to highlight the statistical analysis.

Response:

Thank you very much for this great comment. We have added a subtitle to briefly explain all the statistical methods and techniques used for evaluating the damages caused by the COVID-19 pandemic.

Subtitle: Assessing the ripple effect: using Mann-Whitney U test, histograms, box and violon plots to compare the admissions of different health sectors in the Greater Toronto Area, during the COVID-19 pandemic with before that, and ANOVA and Dunn's test with Bonferroni correction to extract the critical sub-regions

4. The authors could write a legend for each table.

Response:

Great idea! Thank you very much. We have added a legend to all the tables of the manuscript.

5. The sentence " On average, 53.84% (with a 95% CI ranging from a minimum of 53.11% to a maximum of 54.56%), 48.52% (with a 95% CI ranging from a minimum of 46.53% to a maximum of 50.51%), 47.27% (with a 95% CI ranging from a minimum of 46.26% to a maximum of 8.29%), 54.86% (with a 95% CI ranging from a minimum of 53.54% to a maximum of 56.18%), 59.27% (with a 95% CI ranging from a minimum of 58.04% to a maximum of 60.51%), 60.8% (with a 95% CI ranging from a minimum of 59.7% to a maximum of 61.83%), and 61.84% (with a 95% CI ranging from a minimum of 60.97% to a maximum of 62.71%) of patients waiting for surgery, cancer tests, ophthalmology appointments, orthopedic procedures, pediatrics, MRI scans, and CT scans, respectively, have exceeded their waiting time targets." is difficult to understand and should be rephrased; page7, line 53.

Response:

Thank you very much for your concern. We have rewritten this sentence to make it more readable and understandable:

On average, the number of patients that have fallen out of their time targets for surgeries equals to 53.84% (with a 95% CI ranging from a minimum of 53.11% to a maximum of 54.56%), for cancer tests comes to 48.52% (with a 95% CI ranging from a minimum of 46.53% to a maximum of 48.29%), for ophthalmology appointments corresponds to 47.27% (with a 95% CI ranging from a minimum of 53.54% to a maximum of 56.18%), for orthopedic procedures sits at 54.86% (with a 95% CI ranging from a minimum of 58.04% to a maximum of 60.51%), for pediatrics is 59.27% (with a 95% CI ranging from a minimum of 58.04% to a maximum of 60.51%), for MRI scans agrees with 60.8% (with a 95% CI ranging from a minimum of 59.7% to a maximum of 61.83%), and for CT scans matches up with 61.84% (with a 95% CI ranging from a minimum 60.97% to a maximum of 62.71%).

6. " Approximately half of the locations, the number of" should be rephrased, page 9, line 15.

Response:

Thank you very much for your attention. We had forgotten a preposition in that sentence. It is fixed now:

Approximately, in half of the locations, the number of completed procedures for cancer tests, MRI scans, and CT scans increased during the COVID-19 pandemic

7. In Discussion the authors mention that the number of CT scans decreased. I suggest to debate more on this affirmation, as the number of lung CT scans was influenced by the need to classify the lung injury in patients diagnosed with COVID-19 and angio-CT was frequently used to confirm or inform a suspicion of pulmonary embolism, with a high incidence in hospitalized, sever COVID-19 cases, as stated in Tudoran Cristina, Dana Emilia Velimirovici, Delia Mira Berceanu-Vaduva, Maria Rada, Florica Voiță-Mekeres, and Mariana Tudoran. 2022. "Increased Susceptibility for Thromboembolic Events versus High Bleeding Risk Associated with COVID-19" *Microorganisms* 10, no. 9: 1738. <https://doi.org/10.3390/microorganisms10091738>

Response:

Thank you very much for your attention. This is an interesting debate and we have added it to the end of the results section:

Interestingly, the results indicate that although the number of completed CT scans has increased by 2.38% on average during the COVID-19 pandemic compared to before that (Table 5), the number of patients who have surpassed their timeline increased the most for CT scan among all other ISPs, according to Table 4. The reason lies in the high demand for CT scan during the pandemic to classify lung injury in patients diagnosed with COVID-19. Moreover, angio-CT was frequently used during the pandemic to confirm or inform suspicious pulmonary embolism in severe COVID-19 cases [45]. [45] Tudoran C, Velimirovici DE, Berceanu-Vaduva DM, Rada M, Voiță-Mekeres F, Tudoran M, Increased Susceptibility for Thromboembolic Events versus High Bleeding Risk Associated with COVID-19, *Microorganisms*, 2022;10(9) doi: 10.3390/microorganisms10091738.

8. Overall, the Discussion section is too short. The study would benefit if the authors would compare their findings with research from other countries, for example: Pescariu Silviu Alexandru, Cristina Tudoran, Gheorghe Nicusor Pop, Sorin Pescariu, Romulus Zorin Timar, and Mariana Tudoran. 2021. "Impact of COVID-19 Pandemic on the Implantation of Intra-Cardiac Devices in Diabetic and Non-Diabetic Patients in the Western of Romania" *Medicina* 57, no. 5: 441. <https://doi.org/10.3390/medicina57050441>

<https://doi.org/10.3390/medicina57050441>

Response:

Thank you very much for your great comment. We have discussed the generalizability of our work in the discussion and compared our work with other works:

Our findings could be generalized to health services in other locations, since other studies have also found similar pattern of incidences in other countries. Pescariu, et al [46] found that intra-cardiac

device implantation procedures decreased by 75% during the COVID-19 pandemic, in Romania. Qua Quarini, et al [13] reported a significant decline in radiological exams during the COVID-19 pandemic, in Lombardy, Italy. Sutherland, et al [14] found that during the COVID-19 pandemic, breast screening decreased by 51.5%, emergency department visits by 13.9%, and hospital planned surgeries by 32.6% compared to before, in Australia.

Moreover, other studies found similar results in Ontario [16, 19, 20, 23]. However, our work is novel since it has studied the impact of COVID-19 on healthcare system in a more granular scale, i.e., subregions, to extract critical localities that require especial attention for recovery.

[46] Pescariu SA, Tudoran C, Pop GN, Pescariu S, Timar RZ, Tudoran M, Impact of COVID-19 Pandemic on the Implantation of Intra-Cardiac Devices in Diabetic and Non-Diabetic Patients in the Western of Romania, *Medicina (Kaunas)*, 2021;57(5) doi: 10.3390/medicina57050441.

9. Usually Limitations is at the end of Discussions

Response:

Thank you very much for your attention and notification. We have removed the “Strengths and Limitations” and added it to the end of the discussion.

Reviewer: 3

Dr. Makoto Kaneko, Yokohama City University, Hamamatsu University School of Medicine

Comments to the Author:

To the authors

Thank you for providing the opportunity to review the article.

My below comments are based on the STROBE checklist for observational study (<http://www.equator-network.org/reporting-guidelines/strobe/>)

and I hope these comments improve your article.

Major concerns

Title and abstract

#1. Please include the study’s design with a commonly used term in the title or the abstract.

Response:

Thank you very much for this great comment. We have included the study design with common terms in the abstract:

Design: In a **cross-sectional study**, the negative impact of the pandemic on various healthcare sectors, including Preventive and Primary Care (PPC), the Emergency Department (ED), Alternative Level of Care (ALC), and Imaging, Procedures, and Surgeries (IPS) is investigated. **Study questions** include assessing impairments caused by the COVID-19 pandemic and discovering hotspots and critical sub-regions that require especial attention to recover. The **measuring technique** involves comparing the number of cases during the COVID-19 pandemic with before that, and determining the difference in percentage. Statistical analyses (Mann-Whitney U, ANOVA, Dunn tests) is used to evaluate sector-specific changes and interrelationships.

Setting: This work uses **primary data** which was collected by the Black Creek Community Health Center (BCCCHC). The **study population** was from three regions of GTA, namely, the city of Toronto, York, and Peel. For all health sectors, the **sample size** was large enough to have a

statistical power of 0.95 to capture 1% variation in the number of cases during the COVID-19 pandemic compared to before that.

We have also altered the title to include the important parts of the study design:

A cross-sectional study to assess the impact of the COVID-19 pandemic on healthcare services and clinical admissions using statistical analysis and discovering hotspots in three regions of the Greater Toronto Area

The title includes the **type of study** (cross-sectional study), the **questions of the study** (assess the impact of COVID-19 pandemic on healthcare services and clinical admissions and discovering hotspots), the **methods** (which is using statistical analysis), and the **population** (which is in three regions of Greater Toronto Area).

Introduction

#2. The introduction is slightly redundant and the author merely lists up existing research regarding the impact of COVID-19. Thus, please rewrite the section concisely.

Response:

Thank you very much for your helpful comment which remarkably increased the quality of our manuscript. We have made significantly changes to the introduction of the manuscript to explain the problem better, explain some ambiguous parts (e.g. ALC) and remove the redundancy:

The novel coronavirus disease of 2019, known as COVID-19, originated in Wuhan, China, in December 2019, and quickly spread worldwide through international and intra/interstate travel. Shortly thereafter, hospitals and Intensive Care Units (ICUs) were inundated with COVID-19 patients. Many countries implemented lockdown measures to curb the rapid spread of the virus and protect lives. Although Non-Pharmaceutical Interventions (NPI) successfully reduced the number of COVID-19 infections, they had a destructive outcome for the healthcare system [1, 2]. Numerous surgeries, laboratory tests and non-emergency healthcare services were cancelled or postponed. Additionally, people delayed visiting healthcare centers, fearing coronavirus infection, and the associated stigma of testing positive for the virus [3]. Consequently, many patients awaiting for healthcare services have fallen out of their timeline, and need to be sorted.

The COVID-19 pandemic had a devastating impact on patients with chronic conditions and Non-Communicable Diseases (NCD), since they were barely able to meet the medical care that they required [4, 5]. Equally, mental health disorder increased in adults as well as children, and adolescents, whom due to the situation did not get medical treatment, and therefore, need long-term attention to recover [6, 7]. Therefore, it is paramount to identify locations where primary health care and Emergency Department (ED) visits have significantly decreased in them during the COVID-19 pandemic to provide appropriate medication to individuals that require persistent care and attention.

Another concerning issue that occurred during the COVID-19 pandemic is decline in childhood immunization. Disruption in children's vaccination, even for a short period of time, could rise the susceptibility to Vaccine Preventable Diseases (VPD) such as measles, polio, and pertussis, and increase the risk of VPD outbreaks [8, 9]. It is necessary to study this critical matter and provide families with sustained catch-up programs to rectify missing doses.

Despite urgent need to hospital beds, Alternative Level of Care (ALC) dramatically increased in some places, during the COVID-19 pandemic [10, 11]. ALC is the term used to address patients that occupy a hospital bed, but no longer need the intensity of care provided. The most common reasons for the discharge delay in Canada include palliative needs, physical therapy, and requiring transition to another adequate facility [12]. ALC is a costly issue which is considered as a bottleneck to hospital

facilities, especially during epidemics. It is essential to manage and control ALC at its hotspots for higher efficiency in delivering healthcare services, in future outbreaks.

These references were added to the manuscript:

- [1] NG JJ, Ho P, Dharmaraj RB, Wong J, Choong A, The global impact of COVID-19 on vascular surgical services, *J Vasc Surg*, 2020;71(6):2182-2183, doi: 10.1016/j.jvs.2020.03.024.
- [2] Rowe F, Hepworth L, Howard C, Lane S, Orthoptic Services in the UK and Ireland During the COVID-19 Pandemic, *Br Ir Orthopt J*, 2020;15(1):29-37, doi: 10.22599/bioj.153.
- [4] Hebbard C, Lee B, Katare R, Garikipati V, Diabetes, Heart Failure, and COVID-19: An Update, 2021;12:706185, PubMed doi: 10.3389%2Ffphys.2021.706185.
- [5] Clerkin KJ, Fried JA, Raikheikar J, Sayer G, Griffin JM, Masoumi A, et al, COVID-19 and Cardiovascular Disease, *AHA J*, 2020;141(20):1648 PubMed -1655, doi: 10.1161/CIRCULATIONAHA.120.046941.
- [6] Moreno C, Wykes T, Galderisi S, Nordentoft M, Crossley N, Nones N, et al, How mental health care should change as a consequence of the COVID-19 pandemic, *Lancet Psychiatry*, 2020;7(9):813 PubMed -824, doi: 10.1016/S2215-0366(20)30307-2.
- [7] Imran N, Zeshan M, Pervaiz Z, Mental health considerations for children & adolescents in COVID-19 Pandemic, *Pak J Med Sci*, 2020;36(COVID19-S4):S67-S72, doi: 10.12669/pjms.36.COVID19-S4.2759.
- [10] Mathews K, Podlog M, Greenstein J, Cioè-Peña E, Cambria B, Ardolic B, et al, Development and Implementation of an Alternate Care Site During the COVID-19 Pandemic, *Cureus*, 2020;12(10): PubMed e10799, doi: 10.7759%2Fcureus.10799.
- [11] Delgado RC, Quesada PP, García EP, Zabalza IM, Vázquez M, Balín RE, Alternate Care Sites for COVID-19 Patients: Experience from the H144 Hospital of the Health Service of the Principality of Asturias, Spain, *Perhosp Disaster Med*, 2021;36(6):774-781, doi: 10.1017/S1049023X21001102.
- [12] Canadian Institute for Health Information (CIHI), Alternative Level of Care in Canada, Analysis in Brief: Taking health information further, [cited 2024 Feb 7]. Available from: https://publications.gc.ca/collections/collection_2014/icis-cihi/H117-5-30-2009-eng.pdf.
- [8] SeyedAlinaghi S, Karimi A, Mojdeganlou H, Alilou S, Mirghaderi SP, Noori T, et al, Impact of COVID-19 pandemic on routine vaccination coverage of children and adolescents: A systematic review, *Health Sci Rep*. 2022;5(2):e00516, doi: 10.1002%2Fhstr.2.516.
- [9] Ackerson BK, SY LS, Glenn SC, Qian L, Park CH, Riewerts RJ, et al, Pediatric Vaccination During the COVID-19 Pandemic, *Pediatrics*, 2021;148(1):e2020047092, doi: https: 10.1542/peds.2020-047092.

Methods

#3. What does ALC (Alternative Level of Care) mean?

Response:

Thank you very much for asking this question. We mistakenly had not explained this part. This issue has been fixed in the introduction:

Despite urgent need to hospital beds, Alternative Level of Care (ALC) dramatically increased in some places, during the COVID-19 pandemic [10, 11]. ALC is the term used to address patients that occupy a hospital bed, but no longer need the intensity of care provided. The most common reasons for the discharge delay in Canada include palliative needs, physical therapy, and requiring transition to another adequate facility [12]. ALC is a costly issue which is considered as a bottleneck to hospital facilities, especially during epidemics. It is essential to manage and control ALC at its hotspots for higher efficiency in delivering healthcare services, in future outbreaks.

[10] Mathews K, Podlog M, Greenstein J, Cioè-Peña E, Cambria B, Ardolic B, et al, Development and Implementation of an Alternate Care Site During the COVID-19 Pandemic, *Cureus*, 2020;12(10): e10799, doi: 10.7759%2Fcureus.10799.

[11] Delgado RC, Quesada PP, García EP, Zabalza IM, Vázquez M, Balín RE, Alternate Care Sites for COVID-19 Patients: Experience from the H144 Hospital of the Health Service of the Principality of Asturias, Spain, *Perhosp Disaster Med*, 2021;36(6):774-781, doi: 10.1017/S1049023X21001102.

[12] Canadian Institute for Health Information (CIHI), Alternative Level of Care in Canada, Analysis in Brief: Taking health information further, [cited 2024 Feb 7]. Available from: https://publications.gc.ca/collections/collection_2014/icis-cihi/H117-5-30-2009-eng.pdf.

#4. Please describe the study design.

Response:

Thank you very much for this important comment. We have made substantial changes to the methodology of the manuscript to clearly describe the study design [43].

[43] Röhrig B, Prel JB, Blettner M, Study Design in Medical Research, *Dtsch Arztebl Int*, 2009;106(11):184-189, doi: 10.3238%2Farztebl.2009.0184.

We have clearly separated and explained the six aspects of study design [43]: the question to be answered, the study population, the type of study, the unit of observation, the measuring technique, the calculation of sample size:

In the following, all aspects of the study design are thoroughly described [40].

2.1. Questions to be answered

This work is fundamentally beneficial to the policy makers and health officials working at the GTA to understand the damages caused by the COVID-19 pandemic to the health system, plan for recovery, and prepare for future pandemics. Five questions are answered in this work:

1. At the sectors listed below, by what percentage the number of admissions decreased, on average, during the COVID-19 pandemic compared to before that, and where are the hotspots in terms of sub-region?
 - Primary healthcare
 - Childhood immunization
 - Fecal screenings
 - Mammogram screenings
 - Pap smear screenings
 - Substance use
 - Mental health
2. By what percentage has ALC increased, on average during the COVID-19 pandemic compared to before that, and where are the hotspots in terms of sub-regions?
3. For the IPSs listed below, by what percentage the number of patients who exceeded their time target increased on average, during the COVID-19 pandemic compared to before that, and where are the hotspots of this incident in terms of sub-regions?
4. For the IPSs listed below, by what percentage the completed cases decreased on average, during the COVID-19 pandemic compared to before that, and where are the hotspots in terms of sub-regions?
 - Surgeries
 - Cancer procedures
 - Ophthalmic procedures
 - Orthopedic procedures
 - Pediatric procedures
 - MRI imaging

- CT scan
5. By what percentage the number of patients waiting for surgery has increased on average, during the COVID-19 pandemic compared to before that, and where are the hotspots in terms of sub-regions?

2.2. Study Population

The data is concentrated on the York and Peel regions, as well as the city of Toronto, encompassing various sub-regions, including Bramalea, Brampton, Dufferin, East Mississauga, East Toronto, Eastern York Region, Mid-East Toronto, Mid-West Toronto, North Etobicoke Malton West Woodbridge, North Toronto, North York Central, North York West, North West Mississauga, Northern York Region, Scarborough North, Scarborough South, South Etobicoke, South West Mississauga, West Toronto, and Western York Region. The data was collected for 151 different Forward Sortation Areas (FSAs). Figure 1 illustrates the study area, with the pink areas representing the FSAs under investigation. The dashed lines, bold black lines, and bold red lines demarcate the boundaries of the FSAs, sub-regions, and regions, respectively. We have created this map using ArcGis Online [41].

Figure 1: The area under study

Roughly, half of the population are Female (53%). Patients are from all age groups. Table 1 shows the total population in each subregion, and the percentage of people that are 65 years old or above.

2.3. Type of study and unit of observation

This study consumes primary data and is considered as an observational epidemiological study, and a cross-sectional study. The unit of analysis for primary healthcare, cancer screenings (fecal, mammogram, and pap smear), and ED (substance use and mental health) is the number of visits, and for childhood immunization, ALC, and IPS sector (patients who have exceeded their time target, patients who have completed their IPS, and patients waiting for surgery) is the number of patients.

2.4. Measuring technique

The data were collected from various sources, including the Ontario Health Insurance Plan (OHIP), Client and Health Related Information System (CHRIS), Transfer Payment Ontario (TPON), Ontario Drug Benefit Claims (ODBC), Drug and Alcohol Treatment Information System (DATOS), Narcotics Monitoring System, Bed Census Summary (BCS), Ontario Healthcare Financial and Statistics (OHFS), Wait Time Information System (WTIS), Resource Matching and Referral (RM&R), Home and Community Care Support Services, among others.

For certain parameters within the dataset, we compared the number of patients during the COVID-19 pandemic to the period before that for each sub-region, using Equation 1.

$$(1) \quad 100 \times \left(\frac{\text{patients} - \text{during} - \text{COVID} - 19}{\text{patients} - \text{before} - \text{COVID} - 19} - 1 \right)$$

Where, *patients – during – COVID – 19* and *patients – before – COVID – 19* are the number of patients during and before COVID-19 periods, respectively. So, if the number of patients has increased during COVID-19 compared to before that, equation 1 will return a positive and if it has decreased, it will return a negative value.

Missing data which happened only in the PPC sector due to unavailability of the location variable in the OHIP data source, accounted for approximately 12% of the PPC data, and was excluded from the dataset.

The health parameters of each health sector were compared across different sub-regions using histograms, box and violin plots, and through the Mann-Whitney U test. To identify which sub-region is significantly different from the rest, first, ANOVA was used to determine the significance, and then Dunn’s test with Bonferroni correction was applied to identify the exact sub-region(s). P-values lower than 0.05 are considered significant [42]. This study uses the STROBE cross sectional reporting guidelines [43]. Patients or the public were not involved in the design, or reporting, or dissemination plans of this research.

2.5. Calculation of sample size

The sample size of the study was set large enough to have a statistical power of 0.95 for capturing 1% variation in the number of cases, during COVID-19 pandemic compared to before that, for all health parameters under study [44]. Table 1 illustrates the total population size for different sub-regions.

Table 1: The population size and percentage of patients with 65 years of age or over

Sub-region	Population	% With 65 years of age or over
North York West	806894	14.3768%
Eastern York Region	563088	14.2422%
South West Mississauga	459340	15.8967%
East Mississauga	1102439	14.8528%
North West Mississauga	1176836	9.7198%
Brampton	1497688	9.5578%
Bramalea	1251304	10.4156%
North Etobicoke Malton West Woodbridge	547182	13.7505%
Western York Region	896419	13.2478%
Northern York Region	206663	12.0321%
Dufferin	8718	15.7822%
Scarborough South	448819	14.9264%
Scarborough North	181127	19.9786%

East Toronto	322882	13.3525%
North York Central	372532	16.6389%
North Toronto	245466	15.6355%
Mid-East Toronto	178652	11.0301%
Mid-West Toronto	291118	13.028%
West Toronto	277071	13.434%
South Etobicoke	98077	17.415%

Population: actual number of patients who have a profile in the health clinics under study, i.e., Toronto public health unit (master# 3895), York region public health unit (master# 2270), and Peel public health unit (master# 2253).

% With 65 years of age or over: percentage of the population that are 65 years old or above.

Results

#5. Please demonstrate the proportion of missing data among all data

Response:

Thank you very much for your attention. We have reported the proportion:

Missing data which happened only in the PPC sector due to unavailability of the location variable in the OHIP data source, accounted for approximately 12% of the PPC data, and was excluded from the dataset.

#6. The study is a before-after design. Thus, the authors can not investigate causal relationships. This needs to be included in the limitation section.

Response:

Thank you very much for your attention and notification. We have added this to the limitations section.

However, the data were driven only from three health clinics, and from limited number of sources. Moreover, since the study has a before-after design, casual relationships are not captured and cannot be investigated.

Minor concerns

#7. In the Methods section, the authors stated the STROB. It may indicate STROBE.

Response:

Thank you very much for your attention. We have corrected this mistake. (highlighted in yellow).

VERSION 2 – REVIEW

REVIEWER	Tudoran, Cristina University of Medicine and Pharmacy Victor Babes Timisoara, Department VII. Internal Medicine II, Cardiology Clinic I
REVIEW RETURNED	28-Feb-2024

GENERAL COMMENTS	The authors have answered all my questions. Personal note: The re-revision would have been easier to perform if the authors would have marked the changes they made.
---

REVIEWER	Kaneko, Makoto Yokohama City University, Primary Care Research Unit, Graduate School of Health Data Science
REVIEW RETURNED	28-Feb-2024

GENERAL COMMENTS	Thank you for addressing the points which I have raised appropriately.
--